# Unraveling Barriers to a Healthy Lifestyle: Understanding Barriers to Diet and Physical Activity in Patients with Chronic Non-Communicable Diseases

**DOI:** 10.3390/nu15153473

**Published:** 2023-08-05

**Authors:** Massimiliano Cavallo, Giovanni Morgana, Ivan Dozzani, Alessandro Gatti, Matteo Vandoni, Roberto Pippi, Giacomo Pucci, Gaetano Vaudo, Carmine Giuseppe Fanelli

**Affiliations:** 1Department of Medicine and Surgery, University of Perugia, Unit of Internal Medicine, Terni University Hospital, Piazzale Tristano Di Joannuccio, 1, T05100 Terni, Italy; massimilianocavallotr@gmail.com (M.C.); giacomo.pucci@unipg.it (G.P.); gaetano.vaudo@unipg.it (G.V.); 2Post-Graduate School of Sports Medicine, Department of Medicine and Surgery, University of Perugia, 06123 Perugia, Italy; giovamorga94@gmail.com; 3Post-Graduate School of Clinical Nutrition and Dietetics, Department of Medicine and Surgery, University of Perugia, Piazzale Gambuli 1, 06132 Perugia, Italy; dozzaniivan@gmail.com; 4Laboratory of Adapted Motor Activity (LAMA), Department of Public Health, Experimental Medicine and Forensic Science, University of Pavia, 27100 Pavia, Italy; alessandro.gatti08@universitadipavia.it (A.G.); matteo.vandoni@unipv.it (M.V.); 5Healthy Lifestyle Institute, C.U.R.I.A.Mo. (Centro Universitario Ricerca Interdipartimentale Attività Motoria), Department of Medicine and Surgery, University of Perugia, 06126 Perugia, Italy; carmine.fanelli@unipg.it

**Keywords:** barriers, physical activity, healthy nutrition

## Abstract

Although the important contribution of nutrition and physical activity to people’s health is known, it is equally well known that there are many barriers to adherence to healthy habits (i.e., of an organizational, economic, and/or psychological nature) experienced by the general population, as well as by people with non-communicable diseases. Knowledge of these barriers seems essential to the implementation of the activities and strategies needed to overcome them. Here, we aim to highlight the most frequent barriers to nutrition and exercise improvement that patients with chronic-degenerative diseases experience. Drawing from the Pubmed database, our analysis includes quantitative or mixed descriptive studies published within the last 10 years, involving adult participants with non-communicable diseases. Barriers of an organizational nature, as well as those of an environmental, economic, or psychological nature, are reported. The study of patients’ barriers enables healthcare and non-health professionals, stakeholders, and policymakers to propose truly effective solutions that can help both the general population and those with chronic pathologies to adhere to a healthy lifestyle.

## 1. Introduction

It is widely recognized that nutrition and physical activity (PA) are two of the main determinants of maintaining optimal health for the individual [1,2]. Many studies show that following an unbalanced diet [3,4] and being inactive (defined as an accumulation of fewer than 150 min a week of moderate or vigorous PA [5]) are two risk factors for many cardiovascular and chronic degenerative pathologies. However, it is equally well known that many barriers prevent the general population from achieving constant adherence to healthy nutritional and PA habits [6] and that the main barriers encountered are those of an organizational, economic, and/or psychological nature [7]. Indeed, a recent analysis found that more than 27% of the world global population did not reach the suggested guidelines for PA [8], and as reported by Liu et al. [9], more than 85% of American adults consume junk food daily. Several nutrition and PA strategies have been adopted internationally to encourage the achievement of a healthier lifestyle [10,11]. While these issues may initially appear to be primarily related only to healthy adults, they are equally significant in patients with chronic non-communicable diseases (NCDs). In fact, for individuals with NCDs, proper dietary habits together with regular engagement in PA represent key non-pharmacologic therapeutic strategies [10,12]. A well-balanced diet and regular PA support optimal bodily functions, strengthen the immune system, and promote overall well-being [13,14,15]. Additionally, they help control blood sugar levels, manage weight, reduce inflammation, and improve cardiovascular health [15,16,17]. Among well-balanced diet patterns, the Mediterranean diet (MED) has been proposed as an ideal nutritional model against cardiovascular problems [17,18]. MED is considered the gold standard for treatment for preventing and treating NCDs such as diabetes [19], some types of cancer [20], obesity [21,22,23], and neurodegenerative pathologies in old people [24,25]. Moreover, a positive association between adherence to the Mediterranean diet (MED) and muscle strength was observed in the elderly [16]. MED, as shown by Martínez-González et al. [17], has been also strongly associated with a reduced risk of developing coronary heart disease and ischemic stroke, while also promoting improved cardiovascular health.

More in detail, MED is a nutritional model characterized by a balanced combination of seasonal fruit and vegetables, fish, whole grains, legumes, and extra virgin olive oil, with moderate consumption of white meat, dairy products, and red wine [26].

Concurrently promoting the MED and PA is likely to provide an opportunity for metabolic risk reduction [27] and is a strategic key to both prevent and control the development of NCDs [28]. Therefore, understanding and identifying the barriers to a healthy diet and regular PA becomes imperative in order to develop effective interventions to overcome them.

For these reasons, our study aims to highlight the barriers that patients with chronic-degenerative diseases experience in implementing a healthier diet and an exercise-based therapeutic program, by using a quantitative approach to develop a point-by-point list of the most frequent barriers to both nutritional improvement and exercise. By recognizing and addressing these barriers, it is possible to empower patients with NCDs to adopt healthier lifestyles, enhance their quality of life (QoL), and reduce the burden of NCD management. In the last ten years, there has been a significant advancement in the treatment of patients with NCDs. To ensure that we have the latest and most relevant information, we analyzed only studies published between 2013 and 2023. We decided to consider this period with the aim of providing a recent cross-section of the barriers experienced by the world population, to provide ideas for solving current problems. Moreover, we choose to consider adult people because they conduct very similar lifestyles with the aim of making the population homogeneous.

## 2. Materials and Methods

This study used the PRISMA guideline and methodology for systematic review [19]. This review was not registered.

### Search Strategy

The search for studies was conducted by drawing from the Pubmed database using the following search string “(physical activity OR exercise OR diet) (prescription OR participation) AND barriers NOT rehabilitation”. To be eligible, studies must have the following characteristics:

Quantitative or mixed descriptive studies.

Adult population with chronic pathologies of a non-neurological or psychiatric nature.

Interventions of a non-rehabilitative nature.

Period of publication not exceeding 10 years.

The search produced 432 results (Figure 1). The selection of the studies was performed firstly by reading the titles, then by reading the abstracts and excluding duplicates, and finally, the selection of the remaining studies (Table 1) took place with the aid of the Mixed Methods Appraisal Tool (MMAT) version 2018.

## 3. Results

### 3.1. Barriers to a Healthy Diet

The five selected studies enrolled a total of 1855 subjects. These studies are nearly unanimous in indicating as barriers to maintaining adequate compliance with a healthy diet: lack of time, organizational and family problems, the high (or presumed high) cost of a healthy balanced diet, concerns about one’s body/figure/physical health, and motivational difficulty in changing one’s lifestyle.

Gender-related differences. Excepting from the work of Clark-Cutaia et al. [38] in which women represent 37% of the studied population with a limited cohort not letting a deep comparison between male and female, in all the other examined studies the majority of the involved population is composed by women (100% in Bernard-Davila et al. [39] consisting in breast cancer Hispanic survivors; in Mofleh et al. [41], 97.7% of a population of educators from USA; 87% in Mendonça et al. [40] from the Health Academy Program in Brazil; 58% in Miller et al. [37] consisting in breast cancer survivors). Among women, as a synthetic result of all the analyzed studies, differences in the encountered barriers are especially linked to own’s body perception, grade of education, being employed, need to take care of families, and economic concerns. Being younger and with lower income, for example, represents a risk factor for food insecurity in Mofleh et at. [41].

Ethnicity-related differences. In Clark-Cutaia et al. [38] the studied population is quite equally divided between Caucasian and African-American but the sample is too little to understand differences in terms of ethnicity. Bernard-Davila et al. [39] choose Hispanic people according to their being the main minority in the US but, at the same time, one of the less involved communities in clinical studies. In her work, Spanish monolingualism is one of the main determinants of lack of enrollment in the protocol together with being unemployed and considering the study cost-consuming. Ethnicity seems to not represent a statistically significant element in terms of adherence to a nutritional intervention with or without in-person counseling in Miller et al. [37]. In Mofleh et al. [41], being white or not does not represent a statistically significant condition for food insecurity in a population of educators from Pennsylvania.

Lack of time. Clark-Cutaia et al. [38] enrolled 30 patients (mean age 63.3 ± 13.3 years) with end-stage renal failure on hemodialysis replacement therapy who underwent semi-structured telephone interviews to understand the barriers to following the recommended diet recognized as assisting substitution therapy in the treatment of the underlying disease. Among the main barriers indicated were those related to the time required. The long dialysis sessions, which included the transfer time between home and the dialysis center, led patients to skip a main meal with consequent difficulty in preparing healthy meals and a reduced desire to consume them.

In the work of Bernard-Davila et al. [39], albeit indirectly, it emerges that having a job represents a barrier to adherence to a dietary improvement intervention by a group of Hispanic breast cancer survivors (about 38.2% agreement). Similar factors can also be deduced from the work of Mendonça et al. [40] who also adds the time necessary to look after the family as another time constraint. Moreover, in the study by Mendonça et al. [40], although adherence to the intervention implemented at the Brazilian Primary Health Care service was high (possibly due to subject-related factors, the research scenario [health care service: Health Academy Program], the intervention methodology, and/or the bond-building among participants, professionals and the research team), adherence did seem to have been reduced due to 15 issues related to the incompatibility of situations that included work, self-care, care for the other(s), and service working hours (morning).

In contrast, Farmer Miller et al. [37] showed that for cancer survivors an eight-week course of nutrition education, with guaranteed face-to-face psychological support, was more effective, even though more time-consuming than the same intervention carried out electronically with printed material.

Environmental barriers. In the work of Clark-Cutaia et al. [38], the distance from the hemodialysis site combined with the need to utilize various means of transportation represented a further barrier to maintaining adequate adherence to the recommended diet.

Mendonça et al. [40] emphasize in their paper that an adequate setting for nutritional visits is a factor that promotes adherence.

Reading Miller’s [37] work from another point of view, we can see how face-to-face interaction represents a fundamental motivational factor in increasing adherence to a healthy diet in a group of cancer survivors.

Bernard-Davila et al. [39] recognized the monolingualism of Hispanic cancer survivors in an English-speaking setting as a barrier to participation in the dietary intervention envisaged by the study.

Economic and financial barriers. Clark-Cutaia et al. [38] point out that approximately 50% of their enrolled patients said they did not have adequate finances to ensure the maintenance of a suitable diet for end-stage renal failure. Patients reported that dietary recommendations were expensive both in terms of the quantity and quality of the foods recommended. Another common patient concern was the greater ease of obtaining groceries at large stores with the possibility of finding discounted or less expensive items more easily. Some said that they sometimes had to give up drugs due to economic constraints, consequently making food choices immediately subject to direct economic sustainability.

In the work of Mofleh et al. [41], it emerged that in a population of educators of school-age children, the greatest degree of food insecurity manifested itself in younger subjects (31.5% of the population examined), with BMIs compatible with grade I obesity, with fewer years of teaching, and with lower salary levels.

Bernard Davila et al. [39], acknowledge unemployment and concern about the possible costs of the protocol as barriers to enrollment in the nutritional program provided for Hispanic women who survived breast cancer (about 64.3% strongly agreed that it would cost too much).

In contrast, Mendonça et al. [40] in their study aimed to improve fruit and vegetable consumption, found that instead of economic factors, work, having to care for others, and self-care were greater barriers to fruit and vegetable consumption.

Health status. Clark-Cutaia et al. [38] showed that many patients with end-stage renal failure considered their disease status and its consequences (hemodialysis with its timing and related distress) as a barrier to adherence to the best diet regimen.

A reported higher BMI (32.4 mean) in statistically significant terms characterized educators with greater food insecurity in the study by Mofleh et al. [41].

A lower degree of satisfaction with one’s body image was a factor significantly related to lower adherence to the nutritional intervention proposed in the work of Mendonça et al. [40]. This was regardless of the general state of health (presence of metabolic and cardiovascular comorbidities).

Among the factors that affected participation in the nutritional intervention planned by the group of Bernard-Davila et al. [39], was concerned about one’s own health status and degree of disease (Hispanic breast cancer survivor patients). A total of 37% of the participants thought that a dietary intervention could lead to side effects that doctors cannot predict. On the other hand, perceived health status was not a factor significantly correlated with the greater success of an in-person versus remote nutritional education protocol in the cancer survivor population in the work of Miller et al. [37].

Psychological barriers/insights. In the work of Clark-Cutaia et al. [38], hemodialysis patients reported that poor personalization of dietary intervention as a set of general rules to follow represents a barrier to adherence to the intended diet.

An adherence of more than one year to the nutritional intervention outlined in the work of Mendonça et al. [40] was correlated with better outcomes indirectly, how the proactive attitude towards lifestyle change is fundamental.

Bernard-Davila et al. [39] found that improved confidence in the positive outcome of the research appears to be an inducing factor in participation in the nutritional intervention.

### 3.2. Barriers to Exercise

Out of a total population of 1560 subjects tested through a questionnaire regarding the barriers encountered in undertaking physical exercise, 339 declared that they did not have enough time to dedicate to it (22.3%). Another very frequent problem cited was the distance from the gym or unfavorable weather conditions (8.63%, *n* = 131). A total of 242 attributed their state of health as an obstacle to embarking on a healthier life (15.95%). Of these 242, 123 belong to a very fragile category of patients who were undergoing dialysis therapy; a therapy that very often leads to symptoms related to hypotension, fever, joint pain, and fatigue. Furthermore, 33% of the patients involved in the studies had to interrupt the activity due to problems that required hospitalization. A total of 107 complained of mood disorders and showed little motivation to undertake a course in PA (7.05%). Finally, 71 (5.15%) refused to participate in the study as they did not consider it advantageous for their state of health. 

*Gender-related differences.* Several studies suggested that women with NCDs tend to experience greater barriers to PA than men. For example, Pereira et al. [37] found that women with NCDs were less likely to participate in PA than their male counterparts. Pereira et al. [42] proposed several reasons for this gender disparity, including greater involvement in household tasks such as childcare, meal preparation, and cleaning [43]. In addition, women face other barriers, such as a lack of support from their partners, who may not encourage them to engage in PA, safety concerns, and fear of violence in some places used for PA [44]. In addition, Venditti et al. [32] showed that women with NCDs perceive more barriers to PA and weight loss than men. These barriers include internal cues (such as thoughts and moods), social cues and time management, physical events (such as injury or illness), and challenges related to access and weather conditions.

*Ethnicity-related differences.* The study conducted by Venditti et al. [32], while exploring the primary barriers to PA, showed a strong association between barriers to PA and ethnicity. For example, American Indians viewed “self-control” as a greater challenge to participating in PA, while white adults saw “social cues” as a more crucial barrier. Moreover, Pereira et al. [42] revealed differences in PA levels between non-white and white Brazilian adults with NCDs. Non-white adults were found to be less active than their white counterparts. However, Rosa et al. [34] did not find any differences in PA levels based on skin color among adults undergoing hemodialysis.

*Lack of time.* The questionnaire results allow us to make a series of important conclusions regarding the perception of PA as a therapeutic tool and the barriers that prevent patients from utilizing it. First, it is immediately evident that one of the main motivations preventing patients from undertaking a more active lifestyle is worries about not having enough time available daily [31,32,34,36]. Exercise therapy requires an important weekly commitment with training sessions varying from three to five sessions of approximately 45–60 min each. Finding time to dedicate to PA during the day, especially for people busy with work [31,32,34,36], can, therefore, be extraordinarily complex. It is one of the main problems to consider when prescribing exercise.

In Rosa et al. [34] it emerged that 39.8% of patients did not have enough time for dedicated PA. This seems to be correlated with a sedentary lifestyle, since only 33% of those who already performed recreational PAreferred to lack of time as an effective barrier.

*Environmental barriers.* The difficulty in reaching the site for PA is another frequent type of barrier, which overlaps with the inconveniences of daily life. This also includes physical distance and adverse weather conditions which can discourage less motivated individuals [30,34,35].

*Health status.* Another interesting fact that we found is the low rate of recruitment to some protocols (30% in the case of Rogers et al. [31]; 20.3% in Koutoukidis et al. [36]). This rate was influenced by some characteristics of the population including youthful age, having one or more chronic conditions, and awareness of having a low level of PA [43,44]. Those who declined participation in the protocols mostly claimed to have a sufficient level of weekly activity [34]. This suggests that the suffering or sedentary patient perceives the possibility that movement can benefit health. In the study conducted by Sheshadri et al. [30], 73% of participants expressed willingness to participate in the protocol, believing that they could obtain a benefit of some kind that was related to QoL (30%) or be able to achieve a condition such that they could meet the criteria for renal transplantation (27%).

In cancer patients, the perception of pain and fatigue, also related to therapies, affected more than half of the cases undertaking an exercise-based protocol [30].

The results also show that deterrents to undertaking a therapeutic path based on exercise strongly depend on the type of disease involved, since, in the context of chronic pathologies such as diabetes or hypertension, lack of time or access to facilities limits those subjects [45], while in the contexts of more complex pathologies, the most prevalent deterrents are concerns for one’s health and low motivation [30,44,45].

Sheshadri et al. [30] found that older age and a higher BMI were associated with increased motivational barriers related particularly to the presence of symptoms.

*Psychological barriers/insights.* From the data analyzed by Venditti et al. [32], another issue emerges in the maintenance of optimal levels of PA after the end of the experimental protocols. The six-month protocol completion rate by participants was an impressive 96%. In subsequent re-evaluations, however, participants had difficulty maintaining sufficient levels of PA; they reported this was related to poor time management, daily life commitments, and self-monitoring problems that prevented them from recording progress. This helps us understand that searching for and removing barriers is on a continuum that must also include the search for a way to ensure adequate maintenance of fitness levels. Additionally, the problem of a person’s approach to physical exercise seems to be directly proportional to a person’s degree of fitness and familiarity with the type of activity. In fact, in a study conducted by Thomson et al. [33], it emerged that the perceived barriers are greater in those who have had only dietary intervention and have not performed any type of PA. The study demonstrated that the extent of these barriers decreased over time only in the group that performed PA, while the group with only nutritional intervention did not show significant variation.

An interesting fact in Rosa et al. [34] is the degree of satisfaction derived from the exercises performed. Approximately 25.5% of the population involved in the study reported as a barrier the lack of propensity to perform certain exercises which could demotivate specific populations (such as people undergoing hemodialytic treatment) from following a therapeutic PA program.

*Communication errors between Clinicians and Patients.* Doctor–patient communication is perhaps the most important component of a treatment regimen [46]. A patient must trust the doctor and be well-informed by the doctor; a trusting, well-informed relationship often leads the person needing treatment to follow the doctor’s advice to the letter. It is not surprising that if patients with a chronic-degenerative disease such as diabetes, are not educated to have a healthy awareness and acceptance of the condition through a broad and exhaustive informative interview, some may find themselves hesitant about what to do to deal with it. The need to develop a healthier lifestyle is often a topic dealt with superficially by the treating doctor, with little specific direction given as to methods for doing so.

## 4. Discussion

It should be noted that in literature it is easier to find qualitative rather than quantitative studies that evaluate the barriers to acquiring a healthier lifestyle through an adequate diet and regular physical activities. This is one of the peculiarities that makes this literary review innovative. In this discussion, we compare the result of our review of quantitative mixed studies and qualitative ones.

Patients suffering from chronic NCDs experience a condition of quoad vitam persistence of their morbid condition. The indefiniteness of the duration of the disease makes them particularly exposed to the risk of not adhering to recommended treatment, even in the context of a therapeutic alliance between doctor and patient. Although it is widely known that a healthy diet and maintenance of an adequate level of PA represent a fundamental component of the management and improvement of cardiovascular health [17] and most chronic NCDs in both clinical terms and QoL, it is equally commonly known that patients with NCDs experience numerous barriers to maintaining these prescriptions. Regarding adherence to recommendations, some elements must be considered. There could be an impact depending on the region of origin of the authors’ study and the patients involved [17]. For example, many of the investigators who supported MED have born or lived in Mediterranean countries and this could have contributed to the adoption of their opinions on the benefits of MED [47]. This potential bias is not supported by the results of solid studies conducted in non-Mediterranean populations that have found similar benefits in adhering to MED [17]. Similarly, it was observed that adherence to WHO PA guidelines was lower among adults from Southern and Central European countries (Romania, Poland, Croatia, Cyprus, and Malta) and the USA than among Northern European countries (Iceland, Sweden, The Netherlands, and Denmark). Moreover, PA adherence was higher in men than in women [48]. Furthermore, our findings highlighted the presence of ethnicity and gender differences that frequently limit the participation of adults with NCDs in PA.

In our literature review, various similar barriers to healthy diet and exercise emerged. Among these, first, the time needed to be devoted to these two determinants of health is often considered unsustainable, especially for those who are engaged in work activities, who are caring for the family, or for those for whom the chronic disease itself involves considerable consumption of time (i.e., hemodialysis for chronic renal failure, time for therapy, and visits in cancer).

These results appear in agreement with that from qualitative research about these barriers. In Suderman et al. [49], for example, the lack of time is a barrier that remains both at the beginning of the intervention and at the end of it. In particular, workers, that tend to be younger than patients with NCDs seem to encounter this barrier more often. Considering that the number of younger people affected by NCDs is constantly growing in the last decades, especially due to the more effective screening campaign, the more effective diagnostic tools, and the greater sensitivity to their own health, it is particularly important to know this barrier in order to offer useful solutions to overcome them mitigating the severity of disease, especially in a pre-clinical phase or in an early stage in people employed in working activities. Taking care of their own family represents, also in qualitative studies, such as that of Attwood et al. [50] and Sebire et al. [51], an important barrier.

The importance of personalized counselling aimed at maintaining high motivation to achieve appropriate levels of PA and a balanced diet is consistently recognized as a crucial requirement for a broad range of individuals with chronic diseases. Even the accessibility of the exercise or dietary counselling site appears to be a vital element in increasing adherence to behavioral prescriptions. Economic barriers seem to detract more from maintaining adequate dietary levels. Moreover, the perception of a higher cost and a greater effort in procuring food hinders adherence to dietary recommendations. One’s own perception of one’s health and disease are a common barrier to diet and exercise. Patients with a negative subjective sense of well-being have more difficulty maintaining optimal levels of behavioral prescriptions. A close relationship between therapists and patients affected by NCDs, in terms of engagement and therapeutic alliance, appears to be a factor that can promote adherence to dietary recommendations and implementation of PA. In particular, an in-person individualized approach with repeated adherence assessments seems to represent the most effective strategy for supporting and strengthening the motivation to change [52,53]. Knowledge of the specific and common barriers to two key behavioral prescriptions (diet and PA) offers a fundamental opportunity to develop concrete approaches, individual and group, aimed at overcoming these barriers to improvement of the QoL and long-term control of NCDs.

All of these data seem to be similar to that from qualitative studies. In fact, among barriers frequently encountered in qualitative studies, consistent with the results of the review, there are those of an environmental nature and in particular, as noted by Suderman et al. [49], the fear of not finding competent personnel who are able to manage training programs for patients with chronic diseases. In this case, the issue concerns not only the problem of training competent personnel in the matter but also the scarce dissemination of information on the matter and the little trust that patients place both in the type of intervention and in who should be responsible for accompanying and supporting the patient along the course of care. Finally, the note reported by Suderman et al. [49] is noteworthy, in relation to the judgment of training programs, often considered punitive by the patients themselves. In Attwood et al. [50], instead, a good part of patients perceive such programs as constraints to change their lifestyle.

Comparing results from our literary review and data from qualitative studies, more similarities than differences could be appreciated. Due to the seminal importance of keeping an adequate diet and physical activity to counteract the ominous effects of NCDs, it is important to keep in mind that a lifestyle intervention of this type has a considerable impact on the patient’s quality of life and for this reason, the prescription must necessarily take into consideration the needs and expectations of the patient himself. Compliance, in this type of patient, is a fundamental element for the success of the intervention, so in our opinion, the moment in which these therapeutic options are proposed is of fundamental importance. The patient must be aware of both the advantages and disadvantages and limitations that such an intervention can represent. The engagement of patients in their therapeutical plan is a decisive moment in the management of people suffering from NCDs and this could not take place without a complete and deep knowledge of the main behavioral barriers.

In relation to gender differences, women tend to report greater barriers, especially towards increasing levels of physical activity and these barriers are more frequently attributable to psychological problems, environmental barriers, and low motivation than men [28]. In dietary intervention, there are no differences in gender. Ethnic differences affect specifically diet-based interventions, and this is due to sociocultural and economic differences. Hispanic and African-American ethnicity, in the US context, do not enjoy high economic possibilities and often find themselves living in communities where mutual support is essential [50]. Clark-Cutaia [38] highlights how the assumption, by these patients, of a different lifestyle, would tend these subjects to exclude themselves from the community in which they live and to lose the social support of friends and relatives.

An increase in the number of barriers to diet has also been reported by Mofleh et al. [41], whose results show that there are differences regarding insecurities in one’s own eating habits precisely in relation to ethnic and cultural differences. In physical activity intervention, there are differences in race too. If we consider the study taken by Venditti et al. [32] regarding non-Caucasian people, there are about 10–20% barriers more than among Caucasian people and this appears to be related to the socio-economic differences and to the difficulty to access the dedicated structure. In Sheshadri et al. [30], instead, there are no differences in gender in a pedometer intervention, this is probably attributed to the specific population (dialysis patients) and the type of intervention that does not require any gym or specific tools to be practiced.

Finally, as reported by Mendonça [40] and Clark-Cutaia [38], in Brazil and Iran, the level of education seems to be a determining factor for the adherence by these subjects to a healthier lifestyle, probably in relation to a degree of higher literacy.

It must be remembered that the time frame examined in this study (last ten years) also includes one of the greatest global health crisis periods in recent history. In fact, in February 2020 the World Health Organization (WHO) declared a public health emergency due to the SARS-CoV-2 virus, which is particularly dangerous for people with NCDs. To avoid the spread of the virus, containment measures were adopted with the ban to get away from their home. A “lockdown” and social isolation were imposed on some world countries, and it led to a drastic alteration of lifestyle habits (i.e., going to the grocery store was much more difficult and it was forbidden to go to eat meals in restaurants, with a consequent increase in meals cooked and eaten at home [54]. Moreover, PA outdoors are stopped, and exercising at home remained the only possibility to stay active during the COVID-19 pandemic [55]. These changes in habits lead to a reduction of PA levels [56] affecting psychological well-being, and worse nutritional habits [57] (i.e., an increase in the number of snacks consumed, an increase in comfort foods, and alcohol consumption). Anyway, during the COVID-19 pandemic were referred some specific barriers to healthy nutrition, such as, i.e., not having sufficient money to buy groceries and worrying about getting COVID-19 at the store [58]. While regarding PA barriers [59], laziness and fatigue, lack of motivation”, and lack of time were the most prevalent, together with a lack of appropriate facilities, equipment, or space.

*Strengths and Limitations.* A considerable strength is that the large group of authors allowed us to carefully evaluate and double-check the entirety of all the articles. In fact, in the first phase, one group dealt with diet and another with physical activity, based on the authors’ degrees and areas of expertise. Subsequently, the articles selected were evaluated by all the authors. This allowed us to better identify some categories of barriers common to both nutritional and PA recommendations.

The most important limitation, in our opinion, is that only one relevant database was searched (PubMed). Secondarily, only articles available in full-text versions were evaluated for inclusion in our review, which could restrict search opportunities.

Finally, the keywords used to study research were very detailed and could have conditioned the open-field search strategies.

## 5. Conclusions

Although it has been a consistent subject of research in recent years, the study of the barriers that prevent patients from adopting a healthy nutritional and PA lifestyle remains a crucial research focus. Greater knowledge of the various types of barriers patients face would allow clinical team professionals who care for patients with NDCs to more accurately understand the obstacles that hinder patients’ adherence to a healthy lifestyle. Early identification of which barriers patients experience would improve awareness of their difficulties and enable healthcare and non-health professionals to propose truly effective solutions. Given the complex nature of some barriers, early identification would allow the various stakeholders to collaborate in the creation of environments, policies, and intervention programs that can help the general population as well as those with chronic pathologies to adhere to a healthy lifestyle, an effective tool for the prevention and treatment of NCDs. Tailoring lifestyle plans is a seminal element to improve NCD patients’ compliance. Every patient, in fact, presents his/her own needs and motivations that turn away from spending time and resources to obtain a healthier lifestyle. All the members of the community could be engaged in social and political interventions having the aim of stimulating awareness of the real efficacy of a healthier lifestyle in improving global quality of life. An imperative role of health professionals has to educate, ameliorating people’s awareness about NCDs and the most effective actions to counteract the ominous effects of them in terms of health and quality of life. Only a real paradigm shift from a simple taking care of health problems to a global approach to the person in terms of motivation, engagement, and deep knowledge of both physical and psychological barriers to a healthy lifestyle could reduce the burden of NCDs and consequent morbidity, mortality, and disability.

## Figures and Tables

**Figure 1 nutrients-15-03473-f001:**
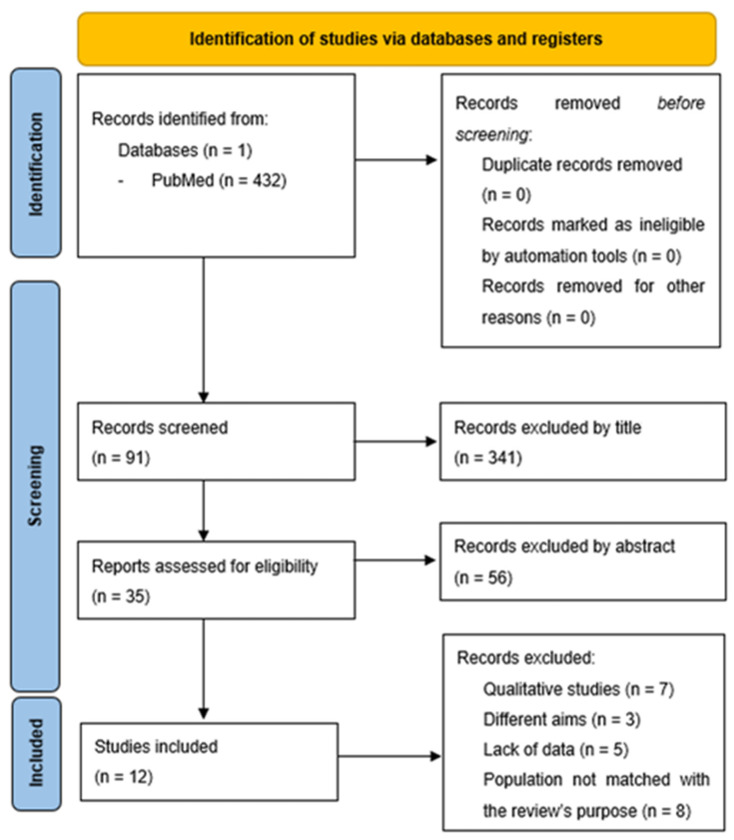
PRISMA [29] 2020 flow diagram for new systematic reviews.

**Table 1 nutrients-15-03473-t001:** Summary of the population characteristics of each study and relative common barriers.

	Population	Gender	Mean Age (% of Population)	Nationality	Ethnicity	Comorbidity	BMI (% of Population)	Common Barriers
Sheshadri 2020 [30]	25	93% M 7% F	60 ± 7	USA	13% white 47% black 20% asian 20% others	93% hypertension 33% and diabetes 37% chronic ischemic heart disease 30% heart failure 7% stroke 13% peripheral arterial disease 20% arrhytmias 100% hemodyalisis	29.1 ± 3.8	Health status
Rogers 2014 [31]	183	53.3% M 46.7% M	62 ± 2 (32.2) 67 ± 2 (32.2) 73 ± 2 (35.5)	Great Britain	100% white	58.5% with 1 or more NCDs 16.8% with musculoskeletal diseases	>25 (59.6)	Environment Health status
Venditti 2014 [32]	1076	31.8% M 68.2% F	34 ± 9 (33.08) 52 ± 7 (45.26) >60 (21.65)	Multicentric International study	53.71% white 18.86% afro-american 16.54% hispanic 5.57% asian	-	33.9	Lack of time Environment Insigths
Thomson 2016 [33]	43	100% F	30.3 ± 6.2	Australia	-	100% PCOS	36.4 ± 5.6	Insights
Rosa 2015 [34]	98	57% M 43% F	51.6 ± 15.7	Brazil	59% white or others 39% black	55% hypertension/diabetes 36% metabolic syndrome 92% cardiovascular diseases 16% musculoskeletal diseases	-	Lack of time Insights
Kang 2022 [35]	52	100% M	63.4 ± 7.1	Canada	89% white 11% others	60% artrhitis 31% hypertension 100% prostate cancer survivors	29 ± 4.7	Environment Health status
Koutoukidis 2017 [36]	83	100% F	62.6 ± 9	Great Britain	-	100% uterine cancer survivors	-	Health status
Miller 2020 [37]	54	100% F	61.2	USA	77% non-hispanic white 8% non-hispanic black 4% hispanic 6% others	100% breast cancer survivors	<18.5 (4) 18.5–24.9 (32) 25–29.9 (25) >30 (40)	Environment Insights Health status
Clark-Cutaia 2018 [38]	30	63% M 37% F	<65 (47) >65 (53)	USA	53% white 47% afro-american	100% haemodyalisis 60% cardiovascular diseases	-	Economic and financial Environment Health status Insights
Bernard-Davilla 2015 [39]	102	100% F	56.4 ± 9.6	South America	25.7% black 40% white 2.9% native americans 15.7% others	100% breast cancer survivors	-	Lack of time Environment Economic and financial Health status Insights
Mendonça 2019 [40]	1483	12.94% M 87.05% F	30 ± 9 (9.8) 50 ± 9 (47.13) >60 (43.08)	Brazil	-	53.67% hypertension 43.42% dyslipidemia 8.5% diabetes	-	Lack of time Environment Economic and financial Health status Insights
Mofleh 2021 [41]	216	2.3% M 97.7% F	41.1 ± 11.9	USA	78.2% white 21.8% others	100% and metabolic syndrome or obesity	30.1 ± 8	Economic and financial Health status

## Data Availability

Not applicable.

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
