# Peer review of "Unraveling Barriers to a Healthy Lifestyle: Understanding Barriers to Diet and Physical Activity in Patients with Chronic Non-Communicable Diseases"

_nutrients, 2023, doi:10.3390/nu15153473_

Round 1

Reviewer 1 Report

The Paper “Unraveling barriers to a healthy lifestyle: Understanding barriers to diet and physical activity in patients with chronic non-communicable diseases” reports interesting data that the authors extrapolated from research papers from the past 10 years.

The topic regards adult population  with non-communicable diseases. Important environmental, economic and psychological aspects are reported. These data are extremely important in order to raise awareness among specialized personnel for the implementation of projects that aim to safeguard human health.

I suggest minor revision.

-         The authors should better discuss the choice to analyze only the data of the last 10 years.

-         Although the purpose of this review is to highlight the barriers that patients with degenerative diseases experience in implementing a healthier diet and an exercise-based diet, in my opinion the authors should also report some information from the literature with examples of healthy diets. For this reason I suggest to the authors to report some hints related to the Mediterranean diet, emphasizing the benefits, also related to physical activity.

-         The authors report the following paper in refereces: “Martínez-González MA, Gea A, Ruiz-Canela M. The Mediterranean Diet and Cardiovascular Health. Circ Res. 2019 414 Mar;124(5):779-798. doi: 10.1161/CIRCRESAHA.118.313348. PMID: 30817261.” In my opinion, more discussion in the text should be reported.

The study is well detailed and the search strategy is well illustrated.

The figures and tables are quite clear and very explanatory. The conclusions are also consistent with the data and the discussion.

Author Response

Review Report Form (Reviewer 1)

RC=> Comment of Reviewer 1

A=> Author's Reply to the Review Report (Reviewer 1)

Comments and Suggestions for Authors

The Paper “Unraveling barriers to a healthy lifestyle: Understanding barriers to diet and physical activity in patients with chronic non-communicable diseases” reports interesting data that the authors extrapolated from research papers from the past 10 years.

The topic regards adult population  with non-communicable diseases. Important environmental, economic and psychological aspects are reported. These data are extremely important in order to raise awareness among specialized personnel for the implementation of projects that aim to safeguard human health.

A: We thank the reviewer for his/her general appreciation of our work.

RC1: The authors should better discuss the choice to analyze only the data of the last 10 years.

A: We thank the reviewer for the suggestion. We added it at the end of the introduction section. Moreover, we have now modified the text. Please see lines 77-83: “In the last ten years, there has been a significant advancement in the treatment of patients with NCDs. To ensure that we have the latest and most relevant information, we analyzed only studies published between 2013 and 2023. We decided to consider this period with the aim of providing a recent cross-section of the barriers experienced by the world population, to provide ideas for solving current problems. Moreover, we choose to consider adult people because they conduct very similar lifestyles with the aim of making the population homogeneous”.

RC2: Although the purpose of this review is to highlight the barriers that patients with degenerative diseases experience in implementing a healthier diet and an exercise-based diet, in my opinion the authors should also report some information from the literature with examples of healthy diets. For this reason I suggest to the authors to report some hints related to the Mediterranean diet, emphasizing the benefits, also related to physical activity.

A: We thank the reviewer for the comment. We modified the text in accordance with this precious suggestion. Please see lines 54-68: “Among well-balanced diet patterns, the Mediterranean diet (MED) has been proposed as an ideal nutritional model against cardiovascular problems. MED is considered the gold standard for treatment for preventing and treating NCDs such as diabetes, some types of cancer, obesity, and neurodegenerative pathologies in old people. Moreover, a positive association between adherence to the Mediterranean diet (MED) and muscle strength was observed in the elderly. MED, as shown by Martínez-González et al., has been also strongly associated with a reduced risk of developing coronary heart disease and ischemic stroke, while also promoting improved cardiovascular health.

More in detail, MED is a nutritional model characterized by a balanced combination of seasonal fruit and vegetables, fish, whole grains, legumes, and extra virgin olive oil, with moderate consumption of white meat, dairy products, and red wine. Concurrently promoting the MED and PA is likely to provide an opportunity for metabolic risk reduction and is a strategic key to both prevent and control the development of NCDs.

RC3: The authors report the following paper in refereces: “Martínez-González MA, Gea A, Ruiz-Canela M. The Mediterranean Diet and Cardiovascular Health. Circ Res. 2019 414 Mar;124(5):779-798. doi: 10.1161/CIRCRESAHA.118.313348. PMID: 30817261.” In my opinion, more discussion in the text should be reported.

A: Thanks: we have modified the text. Please see lines 55-56: “the Mediterranean diet (MED) has been proposed as an ideal nutritional model against cardiovascular problems”, and lines 60-62: “MED, as shown by Martínez-González et al., has been also strongly associated with a reduced risk of developing coronary heart disease and ischemic stroke, while also promoting improved cardiovascular health”, as well as lines 316-323: “Regarding adherence to recommendations, some elements must be considered. There could be an impact depending on the region of origin of the authors’ study and the patients involved. For example, many of the investigators who supported MED have born or lived in Mediterranean countries and this could have contributed to the adoption of their opinions on the benefits of the MED. This potential bias is not supported by the results of solid studies conducted in non-Mediterranean populations that have found similar benefits in adhering to MED”.

RC4: The study is well detailed and the search strategy is well illustrated.

The figures and tables are quite clear and very explanatory. The conclusions are also consistent with the data and the discussion.

A: We thank the reviewer for his/her general appreciation of our work.

Reviewer 2 Report

This article are well-structured and provide good insights into the study's objectives. With some minor improvements in formatting especially in the tables.  Authors can address some limitations. The strengths lie in the clear presentation of findings and the emphasis on the importance of identifying barriers and developing effective interventions for patients with chronic NCDs.

Author Response

Review Report Form (Reviewer 2)

RC=> Comment of Reviewer 2

A=> Author's Reply to the Review Report (Reviewer 2)

RC: This article are well-structured and provide good insights into the study's objectives. With some minor improvements in formatting especially in the tables.  Authors can address some limitations. The strengths lie in the clear presentation of findings and the emphasis on the importance of identifying barriers and developing effective interventions for patients with chronic NCDs.

A: We thank the reviewer for his/her general appreciation of our work.

We also thank the reviewer for the comments, and we improved the manuscript to meet your and other reviewers' suggestions.

Reviewer 3 Report

This review manuscript entitled “Unraveling barriers to a healthy lifestyle: Understanding barriers to diet and physical activity in patients with chronic non-communicable diseases” by Cavallo et al. trying to summarize the current research progress about barriers to healthy lifestyle in patients with chronic non-communicable diseases. However, the content of the manuscript was superficial and lack of deep understanding and perspective towards this topic. For example, result section, which in the main text of this review, only include reference 20-33, and most of the text are related to references 20, 21, 22, 23. I would advise the authors to read more literatures and provide a better presentation of this topic. And figures are not fit well for the manuscript. For example, Figure 1 shows the authors strategy to obtain related literatures for preparing this review. However, it does not contribute to the topic of this manuscript. Figure 2, I would also advise the authors provide detailed legends to clearly deliver the points.

Author Response

Review Report Form (Reviewer 3)

RC=> Comment of Reviewer 3

A=> Author's Reply to the Review Report (Reviewer 3)

R: Quality of English Language: English language fine. No issues detected

A: We thank the reviewer for his/her comments. The manuscript has also been revised by an English proofreader.

Comments and Suggestions for Authors

This review manuscript entitled “Unraveling barriers to a healthy lifestyle: Understanding barriers to diet and physical activity in patients with chronic non-communicable diseases” by Cavallo et al. trying to summarize the current research progress about barriers to healthy lifestyle in patients with chronic non-communicable diseases.

RC1: However, the content of the manuscript was superficial and lack of deep understanding and perspective towards this topic. For example, result section, which in the main text of this review, only include reference 20-33, and most of the text are related to references 20, 21, 22, 23.

A: Actually, we can understand this point of view, however, we believe that literature regarding this topic is scarce. Nevertheless, to meet your suggestion, we generally improve the paper, also the result section.

RC2: I would advise the authors to read more literatures and provide a better presentation of this topic.

A: As stated in the previous response, we understand your point of view and for this reason, we improved the introduction, results, and discussion sections.

RC3: And figures are not fit well for the manuscript. For example, Figure 1 shows the authors strategy to obtain related literatures for preparing this review. However, it does not contribute to the topic of this manuscript. Figure 2, I would also advise the authors provide detailed legends to clearly deliver the points.

A: We thank the reviewer for the comments, but this is a point of view. Figure 2 was now deleted, and we improved the manuscript to meet your and other reviewers' suggestions.

Reviewer 4 Report

The article is a review of unraveling barriers to a healthy lifestyle by understanding barriers to diet and physical activity in patients (Why only over 25?) with chronic non-communicable diseases. The authors aim to highlight the most common barriers to improving nutrition and exercise experienced by patients with chronic degenerative diseases.
An analysis of descriptive studies published over the past 10 years is carried out by the authors; studies all involving adult participants only with non-communicable diseases. Barriers of an organizational nature as well as those of an environmental, economic or psychological nature are reported by the authors. The authors conclude by the fact that a study of patient barriers thus enables professionals, actors and decision-makers to propose effective solutions that can thus help the general population and people with chronic pathologies to adhere to a healthy lifestyle.

The study proposed here is certainly a topical subject and therefore of interest. The authors focus their study on the last 10 years; it would be nice to increment the references and really target the entire study period.

Questions and comments;

- The study is focused on the last ten years, therefore with the Covid crisis of 2019-2021; no comments from the authors on this impact possibly.

- Isn't there also an impact depending on the regions of origin of the patients.

- Are there not populations that are more sensitive to the recommendations? Similarly, can there be differences in behavior between men and women.

- Table 1 must be improve

- What is the real interest of Figure 2.

Author Response

Review Report Form (Reviewer 4)

RC=> Comment of Reviewer 4

A=> Author's Reply to the Review Report (Reviewer 4)

Comments and Suggestions for Authors

The article is a review of unraveling barriers to a healthy lifestyle by understanding barriers to diet and physical activity in patients (Why only over 25?) with chronic non-communicable diseases. The authors aim to highlight the most common barriers to improving nutrition and exercise experienced by patients with chronic degenerative diseases.

An analysis of descriptive studies published over the past 10 years is carried out by the authors; studies all involving adult participants only with non-communicable diseases. Barriers of an organizational nature as well as those of an environmental, economic or psychological nature are reported by the authors. The authors conclude by the fact that a study of patient barriers thus enables professionals, actors and decision-makers to propose effective solutions that can thus help the general population and people with chronic pathologies to adhere to a healthy lifestyle.

The study proposed here is certainly a topical subject and therefore of interest. The authors focus their study on the last 10 years; it would be nice to increment the references and really target the entire study period.

A: We thank the reviewer for his/her comment. We explained the reasons why we have only analyzed studies of the last 10 years. Please see lines 77-83: “In the last ten years, there has been a significant advancement in the treatment of patients with NCDs. To ensure that we have the latest and most relevant information, we analyzed only studies published between 2013 and 2023. We decided to consider this period with the aim of providing a recent cross-section of the barriers experienced by the world population, to provide ideas for solving current problems. Moreover, we choose to consider adult people because they conduct very similar lifestyles with the aim of making the population homogeneous”.

RC1: The study is focused on the last ten years, therefore with the Covid crisis of 2019-2021; no comments from the authors on this impact possibly.

A: We thank the reviewer for his/her comment. We have now modified the text. Please see lines 435-452.

RC2: Isn't there also an impact depending on the regions of origin of the patients. Are there not populations that are more sensitive to the recommendations? Similarly, can there be differences in behavior between men and women.

A: We thank the reviewer for his/her suggestion that helps us improve the quality of the paper. We have now modified the text. Please see lines 227-245: “Gender-related differences. Several studies suggested that women with NCDs tend to experience greater barriers to PA than men. For example, Pereira et al. found that women with NCDs were less likely to participate in PA than their male counterparts. Pereira et al. proposed several reasons for this gender disparity, including greater involvement in household tasks such as childcare, meal preparation, and cleaning. In addition, women face other barriers, such as a lack of support from their partners, who may not encourage them to engage in PA, safety concerns, and fear of violence in some places used for PA. In addition, Venditti et al. showed that women with NCDs perceive more barriers to PA and weight loss than men. These barriers include internal cues (such as thoughts and moods), social cues and time management, physical events (such as injury or illness), and challenges related to access and weather conditions.

Ethnicity-related differences. The study conducted by Venditti et al. while exploring the primary barriers to PA, showed a strong association between barriers to PA and ethnicity. For example, American Indians viewed "self-control" as a greater challenge to participating in PA, while white adults saw "social cues" as a more crucial barrier. Moreover, Pereira et al. revealed differences in PA levels between non-white and white Brazilian adults with NCDs. Non-white adults were found to be less active than their white counterparts. However, Rosa et al. did not find any differences in PA levels based on skin color among adults undergoing hemodialysis” and lines 339-351: “Regarding adherence to recommendations, some elements must be considered. There could be an impact depending on the region of origin of the authors’ study and the patients involved. For example, many of the investigators who supported MED have born or lived in Mediterranean countries and this could have contributed to the adoption of their opinions on the benefits of MED. This potential bias is not supported by the results of solid studies conducted in non-Mediterranean populations that have found similar benefits in adhering to MED. Similarly, was observed that adherence to WHO PA guidelines was lower among adults from Southern and Central European countries (Romania, Poland, Croatia, Cyprus, and Malta) and the USA than among Northern Euro-pean countries (Iceland, Sweden, The Netherlands, and Denmark). Moreover, PA adherence was higher in men than in women. Furthermore, our findings highlighted the presence of ethnicity and gender differences that frequently limit the participation of adults with NCDs in PA”. Please also see lines 412-434: “In relation to gender differences, women tend to report greater barriers, especially towards increasing levels of physical activity and these barriers are more frequently attributable to psychological problems, environmental barriers, and low motivation than men. In dietary intervention, there are no differences in gender. Ethnic differences affect specifically diet-based interventions, and this is due to sociocultural and economic differences. His-panic and African-American ethnicity, in the US context, do not enjoy high economic possibilities and often find themselves living in communities where mutual support is essential. Clark-Cutaia highlights how the assumption, by these patients, of a different lifestyle, would tend these subjects to exclude themselves from the community in which they live and to lose the social support of friends and relatives.

An increase in the number of barriers to diet has also been reported by Mofleh et al., whose results show that there are differences regarding insecurities in one's own eating habits precisely in relation to ethnic and cultural differences. In physical activity intervention, there are differences in race, too. If we consider the study taken by Venditti et al. regarding non-Caucasian people, there are about 10-20% barriers more than Caucasian people and this appears to be related to the socio-economic differences and to the difficulty to access the dedicated structure. In Sheshadri et al., instead, there are no differences in gender in a pedometer intervention, this is probably attributed to the specific population (dialysis patients) and the type of intervention that doesn’t require any gym or specific tools to be practiced.

Finally, as reported by Mendonça and Clark-Cutaia, in Brazil and Iran, the level of education seems to be a determining factor for the adherence by these subjects to a healthier lifestyle, probably in relation to a degree of higher literacy”.

RC4: Table 1 must be improve.

A: We thank the reviewer for his/her comments. We have now modified table 1.

RC5: What is the real interest of Figure 2.

A: We thank the reviewer for his/her comments. Figure 2 was now deleted.

Round 2

Reviewer 3 Report

N/A

Reviewer 4 Report

Agree to publication 

All suggestions and propositions are revised